# Role of Peroxisome Proliferator-Activated Receptor α-Dependent Mitochondrial Metabolism in Ovarian Cancer Stem Cells

**DOI:** 10.3390/ijms252111760

**Published:** 2024-11-01

**Authors:** Seo Yul Lee, Min Joo Shin, Seong Min Choi, Dae Kyoung Kim, Mee Gyeon Choi, Jun Se Kim, Dong Soo Suh, Jae Ho Kim, Seong Jang Kim

**Affiliations:** 1Department of Physiology, School of Medicine, Pusan National University, Yangsan 50612, Gyeongsangnam-do, Republic of Korea; coa0121@gmail.com (S.Y.L.); shinmj90@gmail.com (M.J.S.); csm6358@gmail.com (S.M.C.); 3il2ok@naver.com (M.G.C.); wnstp93@gmail.com (J.S.K.); 2HiCellTech Inc., Yangsan 50612, Gyeongsangnam-do, Republic of Korea; kyumkiki@gmail.com; 3Department of Obstetrics and Gynecology, School of Medicine, Pusan National University, Yangsan 50612, Gyeongsangnam-do, Republic of Korea; dssuh@pusan.ac.kr; 4Department of Nuclear Medicine, School of Medicine, Pusan National University, Yangsan 50612, Gyeongsangnam-do, Republic of Korea

**Keywords:** ovarian cancer, cancer stem cells, PPARα, metabolism, mitochondria

## Abstract

Peroxisome proliferator-activated receptors (PPARs), including PPAR-α, PPAR-β/δ, and PPAR-γ, are involved in various cellular responses, including metabolism and cell proliferation. Increasing evidence suggests that PPARs are closely associated with tumorigenesis and metastasis. However, the exact role of PPARs in energy metabolism and cancer stem cell (CSC) proliferation remains unclear. This study investigated the role of PPARs in energy metabolism and tumorigenesis in ovarian CSCs. The expression of PPARs and fatty acid consumption as an energy source increased in spheroids derived from A2780 ovarian cancer cells (A2780-SP) compared with their parental cells. GW6471, a PPARα inhibitor, induced apoptosis in A2780-SP. PPARα silencing mediated by small hairpin RNA reduced A2780-SP cell proliferation. Treatment with GW6471 significantly inhibited the respiratory oxygen consumption of A2780-SP cells, with reduced dependency on fatty acids, glucose, and glutamine. In a xenograft tumor transplantation mouse model, intraperitoneal injection of GW6471 inhibited in vivo tumor growth of A2780-SP cells. These results suggest that PPARα plays a vital role in regulating the proliferation and energy metabolism of CSCs by altering mitochondrial activity and that it offers a promising therapeutic target to eradicate CSCs.

## 1. Introduction

Ovarian cancer is the most lethal of all gynecologic cancers. Because of the absence of early symptoms, approximately 70% of ovarian cancer cases are diagnosed at a late stage with a poor prognosis [1,2]. Ovarian cancer is highly heterogeneous and causes various intrinsic genetic alterations associated with different tumor histologies [3]. Furthermore, ovarian cancer therapy remains challenging because of intrinsic or acquired multidrug resistance and frequent recurrence. Cancer stem cells (CSCs) are a small subpopulation of tumor cells, accounting for approximately 0.01–0.1%, and possess renewal properties [4,5,6]. They are associated with tumor recurrence owing to multidrug resistance and recurrence [7]. CSCs exhibit various mechanisms of drug resistance involving adenosine triphosphate (ATP)-binding cassette (ABC) transporters, aldehyde dehydrogenases, DNA repair systems, and signaling pathways [8]. Therefore, CSCs are a major factor in ovarian cancer malignancy, and CSC elimination is a crucial requirement for improving the survival rate by controlling chemoresistance and recurrence [9].

Even in abundant oxygen, cancer cells consume high glucose levels and undergo glycolysis, a phenomenon called the Warburg effect [10]. This effect is characterized by a metabolic shift from oxidative phosphorylation (OXPHOS) to glycolysis to meet the energy demands for sustained proliferation while generating the building blocks necessary for cellular components [11]. Increasing evidence suggests that the energy metabolism of CSCs primarily relies on mitochondria through OXPHOS, and they can also utilize fatty acid oxidation to produce energy [12]. Extracellular fatty acids are transported into cells via CD36 and then undergo β-oxidation in the mitochondria, producing acetyl-CoA, which enters the citric acid cycle for OXPHOS [13]. In ovarian cancer, the lipid-rich microenvironment plays a crucial role in modulating the proliferation, survival, and invasion of cancer cells during metastatic colonization. This highlights the rationale behind the preferential spreading of ovarian cancer cells to a lipid-rich microenvironment [14,15]. During peritoneal metastasis, the ability of epithelial ovarian cancer cells to survive and metastasize is influenced by the de novo lipogenesis and the β-oxidation of fatty acids (FAO), processes driven by the absorption of fatty acids from ascites or the omental environment [14,16]. However, the role of FAO in the energy metabolism and proliferation of ovarian CSCs remains unclear.

Peroxisome proliferator-activated receptors (PPARs) belong to the nuclear hormone receptor superfamily and are ligand-dependent transcription factors [17]. PPARs have three subtypes, PPARα, PPARβ/δ, and PPARγ. They form a heterodimer with nuclear receptors and regulate downstream gene expression by binding to specific DNA sequences, namely peroxisome proliferator response elements, in the promoter after ligand binding [18,19]. PPAR isoforms are known to be involved in various cellular responses, including energy balance, glucose homeostasis, lipid metabolism, and cell proliferation [20]. PPARα is involved in fatty acid catabolism and enhances cellular uptake of fatty acids and the conversion of fatty acid to acyl-CoA derivatives [21]. PPARβ/δ stimulates fatty acid synthesis [22], and PPARγ regulates adipocyte differentiation and improves insulin resistance [18]. PPARs exert a dual function in tumor progression. Cancer types determine the functions of PPARs [23]. For instance, PPARα and PPARγ mainly act as a tumor suppressor and a promoter, respectively, during tumor progression, whereas PPARβ/δ plays a controversial role in tumorigenesis [24]. PPARα is upregulated in various cancers, and modulation of PPARα activity by agonists or antagonists influences both pro- and antitumoral effects [25]. PPARα antagonists such as NXT629, AA452, MK886, and GW6471 suppress tumor growth [26,27]. In contrast, the PPARα agonist WY-14643 promoted liver tumors in rodent models [28]. PPARα was activated in pancreatic and colorectal CSCs in which lipid droplets accumulated, and PPARα inhibition suppressed cancer stemness [29]. However, whether PPARα is involved in energy metabolism and proliferation in ovarian CSCs remains to be clarified.

This study investigated the roles of FAO and PPARs in energy metabolism and mitochondrial OXPHOS. We found that treatment with the PPARα inhibitor GW6471 or PPARα silencing suppressed energy metabolism and cell proliferation in CSCs. To our knowledge, these results are the first to suggest that PPARα can be a target for therapy for ovarian CSCs.

## 2. Results

### 2.1. Expression and Activity of PPARs Are Increased in Ovarian CSCs

As previously reported [30], we isolated spheroid-forming CSCs from the A2780 epithelial ovarian cancer cell line. Compared with their parental A2780 cells, spheroids derived from A2780 cells (A2780-SP) exhibited increased expression of CSC markers, including aldehyde dehydrogenase 1, CD44, SOX2, and ABC transporters (Appendix A). To explore the energy metabolism of CSCs, we performed a fuel-dependency test using the Mito Fuel Flex test kit (Agilent, Santa Clara, CA, USA). The A2780 cells were highly glucose-dependent compared with the A2780-SP cells, whereas the A2780-SP cells exhibited greater dependence on fatty acids and glutamine than their parental A2780 cells (Figure 1A). PPARs are involved in the regulation of energy metabolism and CSC-like properties [26,29,31,32]. Therefore, we examined the expression and transcriptional activity of PPAR subtypes (PPARα, PPARβ/δ, and PPARγ). These messenger ribonucleic acid (mRNA) expression levels of the three PPAR isoforms in the A2780-SP cells were greater than those in the A2780 cells (Figure 1B). Consistently, the A2780-SP cells showed increased PPAR transcriptional activity compared with the A2780 cells, as measured using the PPAR response element reporter (Figure 1C). These results suggest that the expression and activity of PPARs are increased in spheroid-forming ovarian CSCs.

### 2.2. PPARα Plays a Pivotal Role in the Survival of Ovarian CSC

To explore the roles of PPARs in ovarian CSC proliferation, we examined the effects of pharmacological inhibitors against three PPAR isoforms on A2780-SP cell viability. The viability of the A2780-SP cells was attenuated in a dose-dependent manner in the GW6471-treated group. In contrast, treatment with the PPARγ antagonist GW9662 slightly reduced A2780-SP cell viability, and the PPARδ inhibitor GSK0660 had no significant effect on cell viability (Figure 2A). GW6471 treatment induced the cell death of not only A2780-SP cells, but also their parental, non-CSC, A2780 cells. The IC_50_ values of GW6471 for the A2780-SP and A2780 cells were 17.2 and 35.7, respectively (Appendix A). The spheroid-forming ability of the CSCs was decreased in the GW6471-treated group (Figure 2B,C). Furthermore, treating CSCs with GW6471 increased cleaved caspase 3, an apoptosis marker (Figure 2D). These results indicate that the GW647-induced inhibition of PPARα leads to apoptotic cell death in CSCs.

To confirm PPARα involvement in CSC proliferation, we examined the effects of PPARα knockdown on cell proliferation. The PPAR expression was silenced by transduction with a lentivirus containing PPARα-specific small hairpin RNA (shRNA) (sh-PPARα) and a control shRNA (sh-control). The treatment with sh-PPARα reduced the expression of PPARα in the A2780-SP cells (Figure 3A). The PPARα-silenced A2780-SP cells exhibited reduced cell proliferation compared with the shRNA-infected control cells (Figure 3B). Furthermore, the spheroid-forming capacity of the A2780-SP cells was markedly attenuated by PPARα silencing (Figure 3C,D). The GW6471-induced apoptosis of A2780-SP cells was abolished by PPARα knockdown (Appendix A). These results suggest that PPARα plays a key role in the proliferation and spheroid-forming ability of CSCs.

### 2.3. Treatment with GW6471 Inhibits Mitochondrial Metabolism of CSCs

To investigate whether PPARα is involved in regulating energy metabolism in CSCs, we performed real-time cell metabolism analysis using Seahorse equipment to observe the metabolic changes of CSCs in response to treatment with GW6471. Among the analyzed mitochondrial metabolic parameters of the CSCs, basal respiration, maximal respiration, spare respiratory capacity, and ATP production were significantly decreased in the presence of GW6471 (Figure 4A,B). Subsequently, we used the Mito Fuel Flex test kit to determine whether the treatment with GW6471 affected mitochondrial nutrient dependence. Notably, mitochondrial dependence on fatty acids, glucose, and glutamine for basal respiration was reduced in the GW6471-treated group compared with the control group (Figure 4C). These results indicate that treatment of CSCs with GW6471 impairs mitochondrial metabolism.

### 2.4. Treatment with GW6471 Suppresses Tumor Growth in an Ovarian Cancer Xenograft Model

To determine whether PPARα is involved in tumorigenesis, we investigated the effects of treatment with GW6471 on the in vivo tumor growth of A2780-SP cells. We established a mouse xenotransplantation model by subcutaneously injecting CSCs and administrating phosphate-buffered saline (PBS) or GW6471 to tumor-bearing mice. Following treatment with PBS or GW6471 twice a week for 3 weeks, the mice were euthanized, and tumor tissues were collected (Figure 5A,B). The GW6471-treated group showed a significant reduction in tumor volume and weight compared with the PBS-treated group (Figure 5C,D). Although GW6471 effectively reduced tumor growth, no reduction in the physical condition or weight of the mice was observed (Figure 5E). These results showed that PPARα antagonists significantly attenuated CSC proliferation in in vitro and in vivo xenograft models, suggesting that they are potential candidates for targeting CSCs to eliminate malignant tumors.

### 2.5. PPARα Is Associated with Poor Prognosis in Patients with Ovarian Cancer

To clarify the clinical significance of PPARs in patients with ovarian cancer, we investigated whether the expression levels of PPAR genes were associated with overall survival by analyzing The Cancer Genome Atlas (TCGA) dataset [33]. The survival analysis results showed that only *PPARA* expression levels were significantly correlated with overall survival (log-rank *p* = 0.011, hazard ratio (HR) = 1.37) and disease-free survival (log-rank *p* < 0.001, HR = 1.68) (Figure 6A,B). The expression levels of *PPARG* and *PPARD* were not significantly correlated with the overall survival, and *PPARD* had an HR of <1 (Figure 6C–F). Subsequently, we analyzed whether PPAR gene expression levels differed according to patient survival status. Consistent with the previous survival analysis, only *PPARA* expression levels were significantly higher in the poor prognosis group, whereas *PPARG* and *PPARD* expression levels were not significantly different (Appendix A).

## 3. Discussion

This study demonstrated that treatment with the PPARα antagonist GW6471 or silencing of PPARα expression induced cell death and inhibited the spheroid-forming ability of A2780-SP cells. Consistently, treatment with GW6471 has been reported to inhibit cell viability, proliferation, and spheroid formation in breast CSCs by inducing energy imbalance and metabolic stress [32]. Moreover, treatment with GW6471 induced apoptosis in kidney cancer cells [34] and head and neck paraganglioma cells [35]. Several PPARα antagonists, including NXT629, AA452, MK886, and IB66, also suppress tumor growth [25,36,37,38]. PPARα agonists, such as fenofibrate and clofibrate, also show antitumor effects in several cancer types, including ovarian cancer [39,40]. These results suggest that the role of PPARα in tumorigenesis depends on cancer cell metabolism. Overall, this study indicates that the energy metabolism of CSCs is more susceptible to PPARα antagonists than that of non-CSCs.

The metabolic rewiring of CSCs from glycolysis to OXPHOS makes them more efficient at generating ATP and more resilient to microenvironmental pressures, such as nutrient starvation [41]. PPARα serves as a pivotal lipid sensor and major regulator of lipid metabolism by controlling various genes involved in processes such as fatty acid uptake and mitochondrial β-oxidation [42]. By modulating a range of downstream genes, PPARα enhances FAO, providing cancer cells with the necessary energy and precursors to thrive, especially in hypoxic and nutrient-deprived environments [27]. Both lipid and glutamine metabolism play critical roles in supporting cancer cell proliferation and survival [43]. Compared with non-CSCs, metabolites from oxidative glutamine metabolism in CSCs mainly contribute to the TCA cycle [44], suggesting the importance of glutamine metabolism in CSCs. Furthermore, CD133+ pancreatic CSCs have a higher mitochondrial oxygen consumption rate (OCR) compared with non-CSCs [45]. This study demonstrated that OCR and dependency on using fatty acid, glucose, and glutamine were significantly decreased in response to PPARα inhibition. The significant effect of treatment with GW6471 on various aspects of mitochondrial metabolism, such as basal respiration, maximal respiration, spare respiratory capacity, and ATP production, underscores the potential of PPARα as a promising therapeutic target in CSCs. These metabolic disturbances can disrupt the energy production process crucial for cell viability and proliferation and induce metabolic stress, which can trigger apoptosis [46]. Therefore, PPARα can be used as a therapeutic target to exploit the metabolic reprogramming of CSCs.

We demonstrated that intraperitoneal injection of GW6471 inhibited the in vivo tumor growth of ovarian CSCs in a murine xenotransplantation model. Furthermore, the expression level of PPARα was negatively correlated with the overall survival rate of patients with ovarian cancer. Consistently, high PPARα expression was associated with poor prognosis [47]. Furthermore, PPARα expression was shown to be negatively associated with clinicopathological data, including the overall survival rate [48]. Together with the increased PPARα expression in ovarian CSCs, these results suggest that PPARα may be responsible for the poor prognosis in patients with ovarian cancer.

## 4. Materials and Methods

### 4.1. Materials

Fetal bovine serum (FBS) (#16000044), penicillin/streptomycin (#15140122), Neurobasal Plus medium (#A3582901), B27 supplement (#17504044), GlutaMAX™ supplement (#35050061), TrypLE™ Express enzyme (#12605010), and HEPES (#15630080) were purchased from Thermo Fisher Scientific (Waltham, MA, USA). Recombinant human fibroblast growth factor (FGF) basic (#100-18B) was purchased from PeproTech (Cranbury, NJ, USA). RPMI1640 (#LM 011-01) was purchased from WELGENE (Gyeongsan-si, Republic of Korea). Recombinant human epidermal growth factor (EGF) (#236-EG) was purchased from R&D Systems (Minneapolis, MN, USA). PrimeSurface^®^ ultralow attachment dishes (#MS-90900Z) were purchased from S-Bio (Constantine, MI, USA). The jetPRIME transfection reagent (#101000046) was purchased from Polyplus (Illkirch, France). The Dual-Luciferase Reporter Assay System (#E1960) was purchased from Promega (Madison, WI, USA). Nitrocellulose membranes (#10600001) and the enhanced chemiluminescence Western blotting system were purchased from Cytiva (Marlborough, MA, USA). Na_4_P_2_O_7_·10H_2_O, NaF, Na_3_VO_4_, EDTA, Tris-HCl, NaCl, and Triton X-100 were purchased from Sigma-Aldrich (St. Louis, MO, USA).

### 4.2. Cell Culture

Human ovarian cancer cell line A2780 was purchased from the American Type Culture Collection (Manassas, VA, USA) and cultured in RPMI1640 medium supplemented with 10% FBS and 1% penicillin/streptomycin at 37 °C and 5% CO_2_. To isolate the sphere-forming CSC populations from A2780 cells, the cells were cultured in CSC culture medium (Neurobasal Plus medium supplemented with 1× B27 supplement, 10 ng/mL recombinant human FGF basic, 20 ng/mL recombinant human EGF, 1× GlutaMAX™ supplement, 20 mM HEPES, and 1% penicillin/streptomycin) in ultra-low-attachment dishes at 37 °C and 5% CO_2_. The subculture of A2780-SP cells involved dissociation into single cells using TrypLE™ Express enzyme, followed by incubation with fresh CSC culture medium.

### 4.3. Dual-Luciferase Reporter Assay

Cells were transfected with aP2 promoter-firefly luciferase and pRL-CMV plasmid using the jetPRIME transfection reagent according to the standard protocol of the manufacturer. After 48 h, the transfected cells were lysed and processed using the Dual-Luciferase^®^ Reporter Assay System according to the manufacturer’ protocol. The luminescence of the samples was measured using a VICTOR3 Multilabel Plate Reader (PerkinElmer, Shelton, CT, USA). Relative luciferase activity was calculated as follows: normalized luciferase activity = firefly luciferase activity/Renilla luciferase activity. Relative luciferase activity = normalized luciferase activity of CSC/normalized luciferase activity of non-CSC.

### 4.4. Immunoblotting Assay

Cells were washed twice with Hank’s Balanced Salt Solution (HBSS) (WELGENE, LB 003-04) and lysed in lysis buffer (30 mM Na_4_P_2_O_7_·10H_2_O, 20 mM NaF, 1 mM Na_3_VO_4_, 1 mM EDTA, 20 mM Tris-HCl, 10 mM NaCl, 1% Triton X-100; pH 7.4). The cell lysates were centrifuged at 15,000 rpm for 15 min at 4 °C, and the supernatants were used for Western blotting. Cell lysates were resolved by sodium dodecyl–sulfate polyacrylamide gel electrophoresis and transferred to nitrocellulose membranes. After blocking with 5% nonfat milk for 30 min at room temperature, the membranes were incubated with antibodies against ABCB1 (Cell Signaling Technology, Danvers, MA, USA, #12683), ABCG2 (Abcam, Cambridge, UK, #ab3380), SOX2 (Abcam, #ab97959), CD44 (Cell Signaling Technology, #5640), ALDH (BD Biosciences, Franklin Lakes, NJ, USA, #611194), and GAPDH (Santa Cruz Biotechnology, Dallas, TX, USA, #SC-47724) overnight at 4 °C. The membranes were then incubated with horseradish peroxidase-conjugated secondary antibodies for 2 h at room temperature and visualized using the enhanced chemiluminescence Western blotting system (Amersham ImageQuant 800, Cytiva).

### 4.5. Quantitative Reverse Transcription-Polymerase Chain Reaction

RNA extraction was performed using TRIsure™ (Meridian Bioscience, Cincinnati, OH, USA, BIO-38033) according to the standard protocol. Complementary DNA was synthesized from 2 µg of mRNA using the Helix Cript™ Thermo Reverse Transcriptase (with dNTP Mix) product (NanoHelix, Daejeon, Republic of Korea, RT50KN) and RNase inhibitor (NanoHelix, RNI2000) according to the standard manual. Quantitative reverse transcription–polymerase chain reaction (PCR) was performed using an ABI QuantStudio3 (Applied Biosystems, Waltham, MA, USA) with SYBR Green PCR Master Mix (Thermo Fisher Scientific, #4367659) according to the manufacturer’s protocol. *GAPDH* was used as an internal control for quantification of mRNA levels of other genes. The primer pairs were as follows: *GAPDH* (forward: 5′-GGTGAAGGTCGGAGTCAACGGA-3′, reverse: 5′-GAGGGATCTCGCTCCTGGAAGA-3′), *PPARA* (forward: 5′-TCGGCGAGGATAGTTCTGGAAG-3′, reverse: 5′-GACCACAGGATAAGTCACCGAG-3′), *PPARD* (forward: 5′-GGCTTCCACTACGGTGTTCATG-3′, reverse: 5′-CTGGCACTTGTTGCGGTTCTTC-3′), and *PPARG* (forward: 5′-AGCCTGCGAAAGCCTTTTGGTG-3′, reverse: 5′-GGCTTCACATTCAGCAAACCTGG-3′).

### 4.6. Cytotoxicity Assay

To evaluate the effects of PPAR antagonists on CSC viability, A2780-SP cells were placed in ultra-low-attachment 96-well plate (Corning Inc., Corning, NY, USA, 3474) at a density of 1 × 10^4^ cells per well and incubated at 37 °C for 24 h. Cells were then treated with PPARα antagonist GW6471 (Cayman Chemical, Ann Arbor, MI, USA, 11697), the PPARγ antagonist GW9662 (Sigma-Aldrich, M6191), or the PPARδ antagonist GSK0660 (Sigma-Aldrich, G5797) and incubated for 48 h at 37 °C. The 10 μL of 3-(4, 5-dimethylthiazolyl-2)-2, 5-diphenyltetrazolium bromide (MTT) solution (0.5 mg/mL in water, Sigma-Aldrich, M2128) was added into the cells and incubated at 37 °C for 2 h. The resulting formazan crystals were dissolved in 100 μL MTT solvent (10% SDS in 0.01 M HCl) overnight at 37 °C. The absorbance of the solution was determined at 570 nm using Infinite^®^ 200 PRO (Tecan Group Ltd., Männedorf, Switzerland). Cell viability (%) was calculated as follows: Cell viability (%) = Antagonist-treated group absorbance/Control group absorbance.

### 4.7. Spheroid Formation

Cells were plated on ultra-low-attachment dishes at a 5 × 10^3^ cells/dish density. Cells were treated with 0.1% dimethyl sulfoxide (DMSO) or 10 μM GW6471 and incubated at 37 °C for 7 days. After 7 days, the cells were centrifuged at 800 rpm for 2 min, and the supernatant was removed. Cells were resuspended in 100 μL HBSS and carefully transferred to 96-well plates. Bright-field images were then taken using the EVOS M5000 Imaging System (Thermo Fisher Scientific). The spheroids with diameters larger than the indicated values were counted. To compare PPARα knockdown and control cells, both groups were untreated with any pharmacological agent, while all other experimental conditions were maintained as described above.

### 4.8. Immunocytochemistry Staining

CSCs were treated with 0.1% DMSO or 10 μM GW6471 for 48 h. Cells were centrifuged at 2000 rpm for 2 min and the supernatant was removed. Cells were fixed with 4% paraformaldehyde (BIOSESANG, Yongin-si, Republic of Korea, PC2031-100-00) for 40 min and permeabilized with PBS solution containing 0.1% Tween 20 for 10 min. After blocking with 5% BSA for 2 h, cells were incubated with primary antibody against cleaved caspase-3 (Cell Signaling Technology, 9661) overnight at room temperature. Cells were then stained with Alexa flour 488-conjugated secondary antibody (Thermo Fisher Scientific, A-11034) and DAPI (Sigma-Aldrich, D9542). Cells were mounted on the slide glass using ProLong™ Gold Antifade Mountant (Thermo Fisher Scientific, P10144), and images were captured using the confocal microscope system (Carl Zeiss, Oberkochen, Germany, LSM900).

### 4.9. Silencing of PPARα Using shRNA Lentiviral Transduction

Sh-PPARα lentiviral vectors (VectorBuilder, pLV(shRNA)-Puro-U6 > *hPPARA*(shRNA#1)) or sh-control vector (Sigma-Aldrich, SHC001) were cotransfected into HEK293T cells with lentiviral envelope and packaging plasmids (VSV-G and Δ8.9) using jetPRIME transfection reagent. The media containing lentiviral particles were collected 2 days after transfection and filtered using a 0.45 µm filter. The filtered medium was mixed 1:1 with the CSC culture medium, and 10 µg/mL polybrene (Sigma-Aldrich, H9268) was added to treat the CSCs for 24 h. The transduced CSCs were used for experiments after selection with 1 µg/mL puromycin (Sigma-Aldrich, P8833).

### 4.10. Cell Proliferation Assay

CSCs (sh-control and sh-PPARα) were seeded at a density of 1 × 10^4^ cells per well in an ultra-low-attachment 96-well plate. After overnight incubation, we added 10 µL of EZ-Cytox reagent (DoGenBio, Seoul, Republic of Korea, EZ-1000) per well for water-soluble tetrazolium salt assay. The absorbance was determined at 450 nm using the Infinite^®^ 200 PRO (Tecan Group Ltd.) after 24, 48, 72, and 96 h.

### 4.11. Seahorse Assay

The oxygen consumption rate was measured using a Seahorse XFp system (Agilent). Seahorse assays were performed according to the manual (Seahorse XF Assay Learning Center, https://www.agilent.com/en/product/cell-analysis/how-to-run-an-assay, accessed on 9 December 2023). Briefly, for ovarian cancer cells, XFp plates (Agilent, 103022-100) were seeded at 5 × 10^3^ per well and treated with 0.1% DMSO or 5 μM GW6471 for 24 h prior to assay. CSCs were treated with 0.1% DMSO or 5 μM GW6471 for 24 h in low-attachment dishes. Next, 1 × 10^5^ cells/well were transferred to XFp plates coated with Cell-Tak (Corning, 354240) and centrifuged (300× *g*, 1 min). Subsequent steps were the same regardless of the cell type. All assays were performed in XF RPMI medium (Agilent, 103681-100) supplemented with 1 mM pyruvate, 2 mM glutamine, and 10 mM glucose. The XF-calibrant-containing XFp cartridge was hydrated overnight at 37 °C in a CO_2_-free incubator before assay.

The Mito Stress Test was performed using the Mito Stress test kit (Agilent, 103010-100) according to the manual. After loading 1 μM oligomycin, 0.5 μM carbonyl cyanide-4 (trifluoromethoxy) phenylhydrazone, and 1 μM rotenone/antimycin A into the cartridge, the Mito Stress test template was selected in the instrument software, and the analysis was performed. All parameters were calculated using the Seahorse Wave Desktop software version 2.6.3.

The fuel-dependency test was performed using the Mito Fuel Flex test kit (Agilent, 103270-100) according to the manual. After loading 2 μM UK5099, 4 μM etomoxir, and 3 μM BPTES into the cartridge, the fuel-dependency test template in the instrument software was selected, and the analysis was performed. All parameters were calculated using the Seahorse Wave Desktop software version 2.6.3.

### 4.12. Antitumor Efficacy of PPARα Antagonist in a Xenograft Tumor Model

Animal studies were conducted according to the guidelines of the National Institutes of Health Guide for the Care and Use of Laboratory Animals. This study was approved by the Pusan National University Institutional Animal Care and Use Committee (PNU-2024-0413). Six-week-old male BALB/c-nu/nu mice were obtained from Orient Bio Inc. (Seongnam-si, Republic of Korea). The mice were housed under 12 h light/dark cycle conditions, 50−60% relative humidity, and a temperature of 25 °C ± 2 °C with a standard diet in accordance with the guidelines of the Pusan National University pathogen-free animal facility. To establish a murine xenotransplantation tumor model, A2780-SP cells (1 × 10^6^ cells/100 µL PBS with 50% Matrigel) were injected subcutaneously into the right flank of mice. Three weeks after cell transplantation, six mice were randomly divided into two groups (*n* = 3 per group) and were administered 400 µL PBS or GW6471 (20 mg/kg) by intraperitoneal injection into the abdominal cavity twice weekly. During the three-week treatment, the body weight of the mice was measured. The tumor size was measured using a Vernier caliper and calculated using the formula V = 0.52 × length × width × height. All mice were euthanized with CO_2_ gas, and the tissues were collected for tumor weight measurement.

### 4.13. The Cancer Genome Atlas Dataset Analysis

The clinical data and mRNA expression levels of *PPARA*, *PPARG*, and *PPARD* in patients with ovarian cancer were extracted from the TCGA ovarian cancer dataset (2011 version, *n* = 489, and http://cbioportal.org, accessed on 7 May 2024) [33,49]. All analyses of TCGA data were performed using R Statistical Software version 4.4.0 [50]. Survival analysis was performed using the “survival” and “survminer” packages. Patients were stratified into high- and low-expression groups based on the cutoff values for *PPARA*, *PPARG*, and *PPARD* expression levels. The optimal cutoff value for the expression of each gene was determined using maximally selected rank statistics [51], which identified the point of maximum statistical significance in distinguishing between the high- and low-expression groups (Appendix A). Kaplan–Meier survival curves were used to illustrate the survival differences between these groups. HR and associated *p*-values were derived using Cox regression analysis and log-rank tests. For comparison analysis of *PPARA*, *PPARG*, and *PPARD* expression levels between patients with and without events, we divided patients into “living” and “deceased” for overall survival and “disease free” and “recurred/progressed” for disease-free survival. We then tested the significance between groups for each gene using an unpaired *t*-test.

### 4.14. Statistical Analysis

Statistical analyses were performed using SigmaPlot 14 and R Statistical software version 4.4.0 [50]. An unpaired *t*-test was used to compare control and experimental groups. Results are expressed as mean ± standard error of the mean, with *p* < 0.05 considered statistically significant (* *p* < 0.05; ** *p* < 0.01; and *** *p* < 0.001).

## 5. Conclusions

This study demonstrates that pharmacological inhibition of PPARα can effectively inhibit CSC proliferation by interfering with mitochondrial metabolism. These findings open new avenues for therapeutic strategies, particularly for patients with ovarian cancer and high PPARα expression. Nevertheless, additional research is required to clarify the molecular mechanisms involved in the PPARα-mediated regulation of energy metabolism in CSCs.

## Figures and Tables

**Figure 1 ijms-25-11760-f001:**
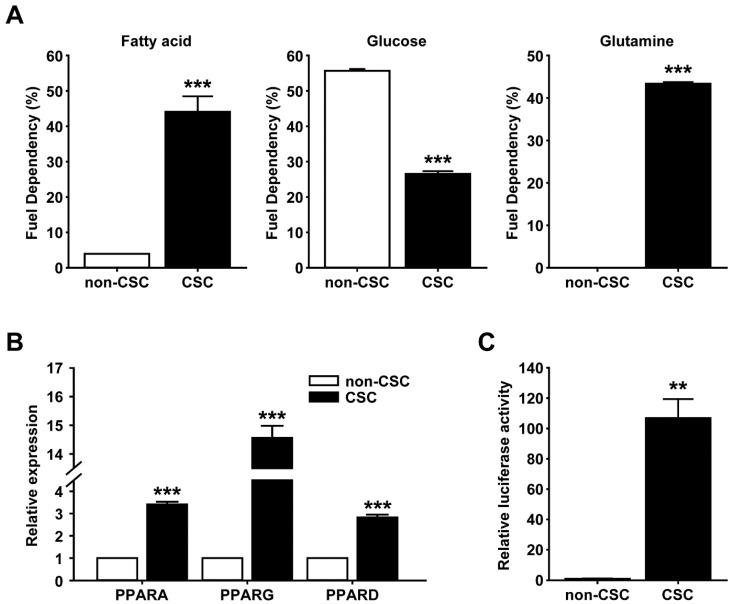
Role of PPARs in energy metabolism of ovarian CSCs. (**A**) The dependence of each carbon source (fuel) dependency was calculated from the oxygen consumption rates measured by the Seahorse analyzer using the Mito Fuel Flex test kit. Cells were treated with 0.1% DMSO or 5 μM GW6471 for 24 h before performing the Mito Fuel Flex test. (**B**) The mRNA expression levels of PPARs in A2780 (non-CSC) and A2780-SP (CSC) cells. (**C**) PPAR transcriptional activity was measured using Dual-Luciferase Reporter assay. Cells were transfected with luciferase plasmid vectors 48 h before measuring luciferase signal activity. Data are presented as mean ± SEM. ** *p* < 0.01; *** *p* < 0.001 (*n* = 3 for each group). PPAR, peroxisome proliferator-activated receptor; DMSO, dimethyl sulfoxide; CSC, cancer stem cell; mRNA, messenger ribonucleic acid; SEM, standard error of the mean.

**Figure 2 ijms-25-11760-f002:**
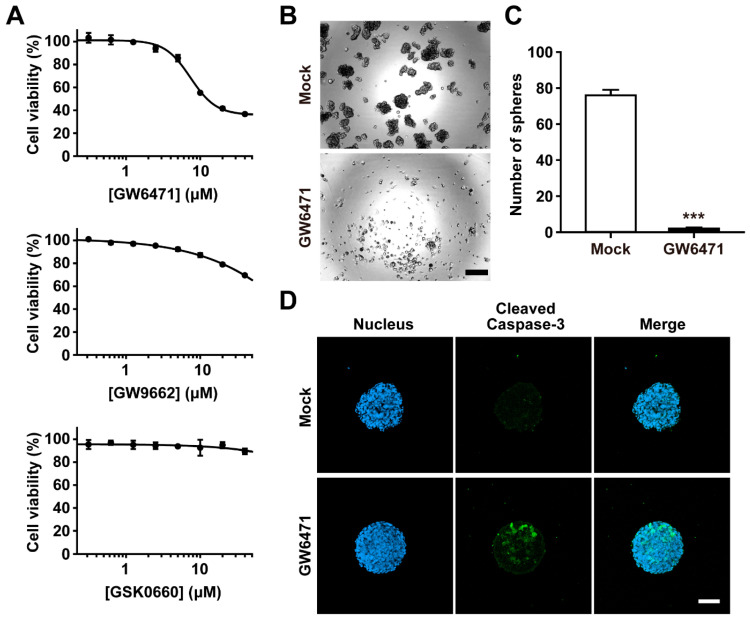
Effects of PPAR antagonists on cell viability of ovarian CSCs. (**A**) Cell viability was determined by MTT assay. CSCs were treated with the indicated concentrations of antagonists (GW6471, GW9662, GSK0660) for 48 h. (**B**) Representative images of spheroid formation assay of CSCs in the presence or absence of GW6471. Scale bar, 200 μm. (**C**) Quantification of spheroid number in spheroid formation assay. The number of spheroids with a diameter > 100 μm was counted. Data are presented as mean ± SEM. *** *p* < 0.001 (*n* = 3 for each group). (**D**) Representative immunocytochemistry images for detection of cleaved caspase-3. CSCs were treated with 0.1% DMSO (mock) or 10 μM GW6471 for 48 h before staining. Scale bar, 100 μm. PPAR, peroxisome proliferator-activated receptor; CSC, cancer stem cell; MTT, 3-(4, 5-dimethylthiazolyl-2)-2, 5-diphenyltetrazolium bromide; DMSO, dimethyl sulfoxide; SEM, standard error of the mean.

**Figure 3 ijms-25-11760-f003:**
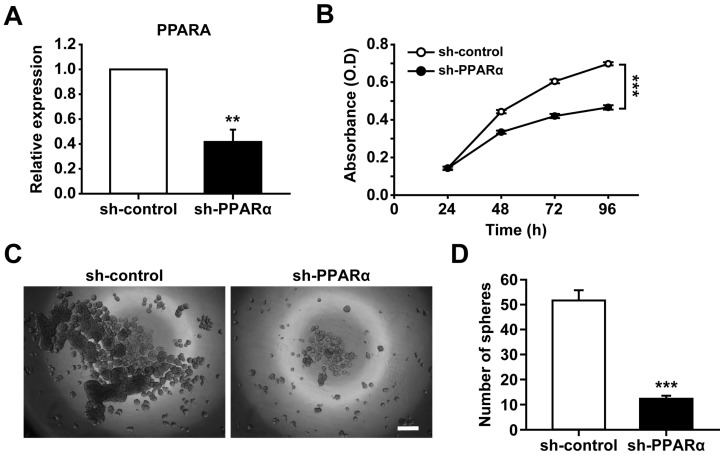
Effects of PPARα knockdown on cell proliferation and spheroid-forming abilities of CSCs. (**A**) Relative mRNA expression of PPARα in CSCs transduced with lentiviruses bearing sh-PPARα or sh-control was measured by qRT-PCR. (**B**) Cell proliferation assessed by WST assay at indicated time points. (**C**) Representative images of spheroid formation in CSCs transduced with lentiviruses bearing sh-PPARα or sh-control. Scale bar, 300 μm (**D**) Quantification of spheroid numbers in spheroid formation assay. The number of spheroids > 150 μm in diameter. Data are presented as mean ± SEM. ** *p* < 0.01; *** *p* < 0.001 (*n* = 3 for each group). PPAR: peroxisome proliferator-activated receptor; CSC: cancer stem cell; mRNA: messenger ribonucleic acid; sh: small hairpin; qRT-PCR: quantitative reverse transcription–polymerase chain reaction; WST: water-soluble tetrazolium salt; SEM: standard error of the mean.

**Figure 4 ijms-25-11760-f004:**
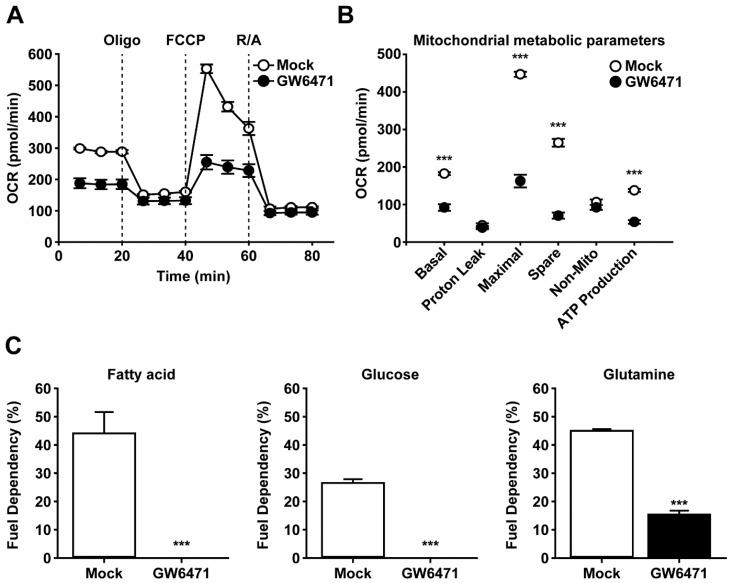
Effects of GW6471 on the mitochondrial metabolism of ovarian CSCs. (**A**) Oxygen consumption rates of the mock and GW6471-treated groups were measured with a Seahorse analyzer using the Mito Stress test kit. Oligo: oligomycin; FCCP: carbonyl cyanide-4 (trifluoromethoxy) phenylhydrazone; R/A: rotenone and antimycin A. (**B**) Mitochondrial metabolic parameters were calculated from the Mito Stress test results. Basal: basal respiration; maximal: maximal respiration; spare: spare respiratory capacity; non-mito: nonmitochondrial respiration. (**C**) Dependence on each carbon source (fuel) was calculated from oxygen consumption rates measured by a Seahorse analyzer using the Mito Fuel Flex test kit. CSCs were treated with 0.1% DMSO or 5 μM GW6471 for 24 h before performing the Mito Stress Test or Mito Fuel Flex test. Data are presented as mean ± SEM. *** *p* < 0.001 (*n* = 3 for each group). CSC: cancer stem cell; DMSO: dimethyl sulfoxide; SEM: standard error of the mean.

**Figure 5 ijms-25-11760-f005:**
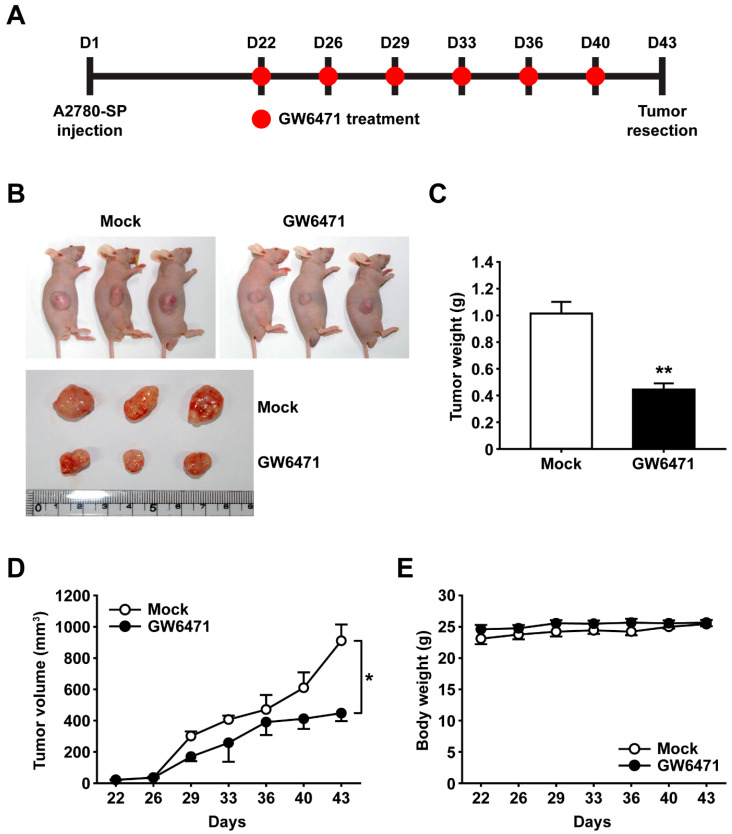
Effects of GW6471 on in vivo tumor growth of ovarian CSCs. (**A**) Experimental overview of ovarian cancer xenotransplantation model. Treatment with PBS or 20 μM GW6471 was started on day 22 after cell injection and continued twice a week until day 40. (**B**) Representative images of mice and resected tumors at day 43. (**C**) Tumor weight measured after resection on day 43. (**D**,**E**) Tumor volume (**D**) and mouse body weight (**E**) were measured twice a week from day 22 to day 43. Data are presented as mean ± SEM. * *p* < 0.05; ** *p* < 0.01 (*n* = 3 for each group). CSC: cancer stem cell; PBS: phosphate-buffered saline; SEM: standard error of the mean.

**Figure 6 ijms-25-11760-f006:**
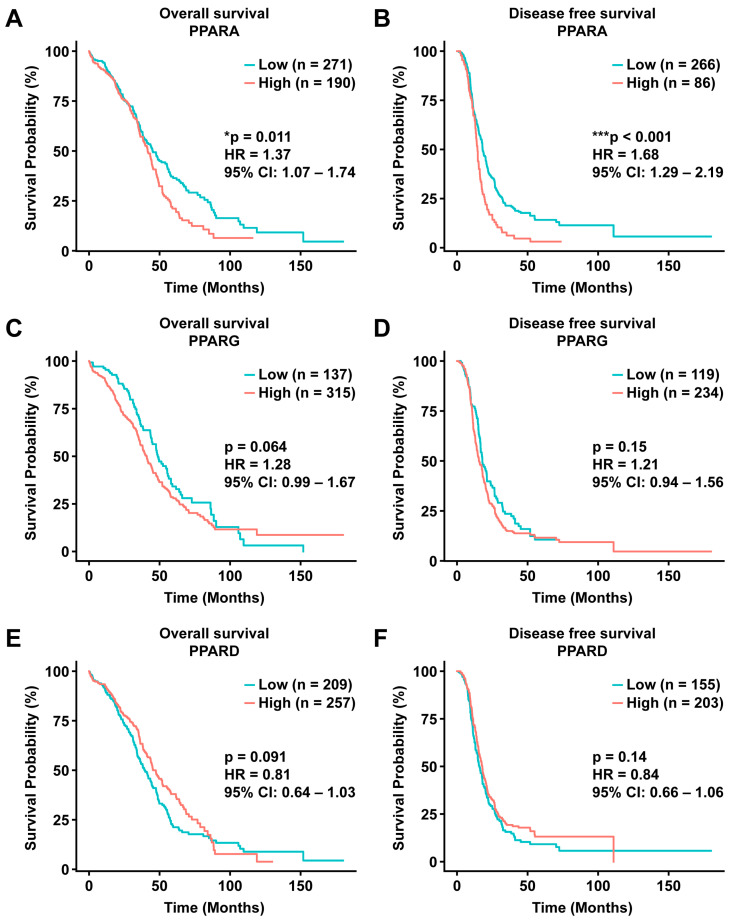
Role of PPARs in the prognosis of patients with ovarian cancer. Kaplan–Meier survival curves for overall survival and disease-free survival stratified by the expression level of PPAR subtypes (*PPARA*, *PPARG*, *PPARD*). (**A**,**C**,**E**) Overall survival of patients with low (blue line) or high (red line) expression of *PPARA*, *PPARG*, or *PPARD*, respectively. (**B**,**D**,**F**) Disease-free survival for patients with low (blue line) or high (red line) expression of *PPARA*, *PPARG*, or *PPARD*, respectively. * *p* < 0.05; *** *p* < 0.001. PPAR: peroxisome proliferator-activated receptor; n: number of patients in each group; HR: hazard ratio; 95% CI: 95% confidence interval.

## Data Availability

The original contributions presented in this study are included in the article/Appendix A. Further inquiries can be directed to the corresponding author/s.

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
