# Peer review of "Role of Peroxisome Proliferator-Activated Receptor α-Dependent Mitochondrial Metabolism in Ovarian Cancer Stem Cells"

_ijms, 2024, doi:10.3390/ijms252111760_

Round 1

Reviewer 1 Report

Comments and Suggestions for Authors

1.      This manuscript is interesting and well-done.

2.      The strength of this article is well organized for readers to understand for regulation of energy metabolism by PGC1a in ovarian CSCs.

3.      Moreover, current study proposed that PGC1a-mediated energy metabolism could be therapeutic approach to cancer stemness of ovarian cancer.

4.      The details about the biological or technical replicates are missing in figure legends.

5.      Line 126, I recommended, ‘Figures 3C and 3D’ > ‘Figure 3C, D’ or ‘Figure 3C and D’ in whole section of Results.

6.      PGC1a known as pivotal regulator of mitochondrial bioenergetics in cancer cell. Current study implied that down modulation of PGC1a was increase of tumor shrinkage (suppression of tumor) in CSCs. Then, in non-CSCs, down modulation of PGC1a is what will happen?

Author Response

Comments 1: This manuscript is interesting and well-done.

Response 1: Thank you for the valuable comments.

Comments 2: The strength of this article is well organized for readers to understand for regulation of energy metabolism by PGC1a in ovarian CSCs.

Response 2: Thank you for the valuable comments.

Comments 3: Moreover, current study proposed that PGC1a-mediated energy metabolism could be therapeutic approach to cancer stemness of ovarian cancer.

Response 3: Thank you for the valuable comments.

Comments 4: The details about the biological or technical replicates are missing in figure legends.

Response 4: Thank you for the helpful comment. According to the referee’s comment, we added the description about the biological replicates in figure legends.

Comments 5: Line 126, I recommended, ‘Figures 3C and 3D’ > ‘Figure 3C, D’ or ‘Figure 3C and D’ in whole section of Results.

Response 5: Thank you for the helpful comment. According to the referee’s comment, we revised the description as recommended in the Results section.

Comments 6: PGC1a known as pivotal regulator of mitochondrial bioenergetics in cancer cell. Current study implied that down modulation of PGC1a was increase of tumor shrinkage (suppression of tumor) in CSCs. Then, in non-CSCs, down modulation of PGC1a is what will happen?

Response 6: Thank you for the valuable question. We found that GW6471 treatment induced cell death of not only A2780-SP cells but also their parental non-CSC, A2780 cells. The IC50 values of GW6471 for A2780-SP and A2780 cells were 17.2 and 35.7, respectively. We described it in Result section and added the results in the Supplementary Figure 2 as follows.

“GW6471 treatment induced cell death of not only A2780-SP but also their parental non-CSC, A2780 cells. The IC50 values of GW6471 for A2780-SP and A2780 cells were 17.2 and 35.7, respectively (Figure S2).”  

Cell viability for GW6471 treatment in non-CSC

IC50 values of GW6471 on cell viability of A2780 and A2780-SP cells 

non-CSC

CSC

IC50

35.7

17.2

95% confidence interval

25.1 – 55.4

14.6 – 20.6

Figure S2. Dose-dependent effects of GW6471 on cell viability of A2780 cells. A2780 cells (non-CSC) were treated with increasing doses of GW6471 for 48 h, and cell viability was measured using MTT assay. The IC50 values of GW6471 on cell viability of A2780 and A2780-SP (CSC) cells.

Reviewer 2 Report

Comments and Suggestions for Authors

The primary goal of the research presented here is to provide evidence to prove that PPARα is a feasible molecular target for removing ovarian cancer stem cells. Selective inhibition of PPARα by a small molecule, GW6471, led to growth inhibition of ovarian cancer stem cells both in cell viability assays and in a mouse xenograft model.  At the cellular level, this was also supported by PPARα knockdown experiments. Given the known functional roles of PPARα in energy metabolism, the authors further tested and confirmed that GW6471 affected mitochondrial metabolism and decreased ATP production, which may account for the inhibition effect of GW6471 on cancer cells. Somewhat similar studies have been done on other cell lines, such as breast cancer stem cells in Ref. 32 and kidney cancer cells in Ref. 34. The experimental methods are solid, and the results support the discussion and the conclusion. The manuscript is well written and easy to follow. Some comments are listed below.

1.  It would be useful to test how GW6471 inhibits ovarian non-CSCs in cell viability assays. Would GW6471 be selective towards CSCs because of overexpression of PPARα?

2.  Based on Figure 1A and Figure 4C, in A2780-SP cells, similar to fatty acids, the demand for glutamine is higher and its metabolism is also negatively impacted by GW6471.  Could the authors clarify why they specifically choose fatty acid in the title of the manuscript?  Is it mainly based on the known functions of PPARα (Ref. 21)?

A minor point:

1.  Please define pLKO.

Author Response

Comments 1: It would be useful to test how GW6471 inhibits ovarian non-CSCs in cell viability assays. Would GW6471 be selective towards CSCs because of overexpression of PPARα?

Response 1: Thank you for the valuable question. We found that GW6471 treatment induced cell death of not only A2780-SP cells but also their parental non-CSC, A2780 cells. The IC50 values of GW6471 for A2780-SP and A2780 cells were 17.2 and 35.7, respectively. We described it in Result section and added the results in the Supplementary Figure 2 as follows.

“GW6471 treatment induced cell death of not only A2780-SP but also their parental non-CSC, A2780 cells. The IC50 values of GW6471 for A2780-SP and A2780 cells were 17.2 and 35.7, respectively (Figure S2).”

Cell viability for GW6471 treatment in non-CSC

IC50 values of GW6471 on cell viability of A2780 and A2780-SP cells 

non-CSC

CSC

IC50

35.7

17.2

95% confidence interval

25.1 – 55.4

14.6 – 20.6

Figure S2. Dose-dependent effects of GW6471 on cell viability of A2780 cells. A2780 cells (non-CSC) were treated with increasing doses of GW6471 for 48 h, and cell viability was measured using MTT assay. The IC50 values of GW6471 on cell viability of A2780 and A2780-SP (CSC) cells.

Comments 2: Based on Figure 1A and Figure 4C, in A2780-SP cells, similar to fatty acids, the demand for glutamine is higher and its metabolism is also negatively impacted by GW6471.  Could the authors clarify why they specifically choose fatty acid in the title of the manuscript?  Is it mainly based on the known functions of PPARα (Ref. 21)?

Response 2: Thank you for the valuable comment. As you mentioned, fuel dependencies of not only fatty acids but also glucose and glutamine were reduced by GW6471 treatment. Therefore, we corrected the manuscript title from "Role of peroxisome-proliferator-activated receptors α-dependent fatty acid metabolism in ovarian cancer stem cells" to "Role of peroxisome-proliferator-activated receptors α-dependent mitochondrial metabolism in ovarian cancer stem cells."

Minor point 1: Please define pLKO.

Minor point response 1: Thank you for the helpful comment. pLKO is an empty vector used as an experimental control in shRNA lentiviral transduction. To clearly describe it, we corrected 'pLKO' to 'sh-control' in the manuscript.